# An integrated model for detecting significant chromatin interactions from high-resolution Hi-C data

Mark Carty[1,2,3], Lee Zamparo[1], Merve Sahin[1,3], Alvaro González[1], Raphael Pelossof[1], Olivier Elemento[2] & Christina S. Leslie[1]

Here we present HiC-DC, a principled method to estimate the statistical significance (P values) of chromatin interactions from Hi-C experiments. HiC-DC uses hurdle negative binomial regression account for systematic sources of variation in Hi-C read counts—for example, distance-dependent random polymer ligation and GC content and mappability bias—and model zero inflation and overdispersion. Applied to high-resolution Hi-C data in a lymphoblastoid cell line, HiC-DC detects significant interactions at the sub-topologically associating domain level, identifying potential structural and regulatory interactions supported by CTCF binding sites, DNase accessibility, and/or active histone marks. CTCF-associated interactions are most strongly enriched in the middle genomic distance range ($\sim$700 kb–1.5 Mb), while interactions involving actively marked DNase accessible elements are enriched both at short ($<$500 kb) and longer ($>$1.5 Mb) genomic distances. There is a striking enrichment of longer-range interactions connecting replication-dependent histone genes on chromosome 6, potentially representing the chromatin architecture at the histone locus body.

[1] Computational Biology Program, Memorial Sloan Kettering Cancer Center, New York, New York 10065, USA. [2] Institute for Computational Biomedicine, Weill Cornell Medical College, New York, New York 10065, USA. [3] Tri-Institutional Training Program in Computational Biology and Medicine, New York, New York 10065, USA. Correspondence and requests for materials should be addressed to C.S.L. (email: cleslie@cbio.mskcc.org).

Hi-C is a genome-wide chromosome conformation capture (3C) technology that uses restriction enzyme digestion of DNA followed by proximity ligation and paired-end sequencing[1]. A large number of paired end reads connecting two genomic regions is interpreted as evidence of an interaction over the population of cells. Hi-C data is typically summarized as a contact matrix relative to a fixed partition of the genome into intervals—at the finest resolution defined by the restriction fragments themselves but more often as longer genomic regions. Each entry or 'interaction bin' in the contact matrix contains the count of unique paired end reads mapping to the corresponding pair of genomic intervals. Not all non-zero counts are significant, and the challenge is to identify interactions supported by more reads than expected by chance. Many factors confound the statistical analysis of Hi-C data. First, random polymer ligation between restriction fragments, which decays as a function of linear genomic distance, produces a background distribution of paired end read counts that likely accounts for a large fraction of Hi-C reads[2–4]. Other systematic sources of bias include GC content and mappability of short reads[5,6]. If uniform genomic bins are used to define the contact map, the number of restriction enzyme sites within each bin is another source of bias.

The original study introducing Hi-C generated a coarse resolution contact matrix (1 Mb bins) to characterize biophysical rules of polymer folding[1], while later studies with improved resolution (10–50 kb bins) reported features of chromatin organization such as topologically associating domains (TADs)—regions that favour internal (within-TAD) contacts over external contacts[3,7]. However, the statistical significance of individual interactions, either within TADs or between more distal loci, was not addressed in these studies. Several normalization schemes have been proposed to correct for GC content and other sources of Hi-C read count bias, including a non-parametric probabilistic approach due to Yaffe and Tanay[6] and iterative correction and eigenvalue decomposition (ICE), which approximates this method[5]. More recently, HiCNorm[8] was introduced to learn these biases statistically with Poisson regression, using GC content and other features as covariates in a generalized linear model (GLM) for interaction bin counts. These normalization approaches are typically used to correct the contact matrix—for example, by rescaling the observed counts in ICE or by replacing the count with the residual from the HiCNorm regression model—to increase reproducibility between experiments for downstream analyses.

Recently several groups have proposed statistical models to assess the significance of interactions—that is, to assign $P$ values to the observed counts in individual interaction bins—for Hi-C and other 3C-based technologies[9,10]. Fit-Hi-C[9] uses a binomial null model $Bin(P(d))$, where $P(d)$ is the probability that a randomly chosen paired end read occurs between a given locus pair at distance $d$. The probability $P(d)$ is estimated from the data using a spline fitting process. To account for other sources of bias, Fit-Hi-C uses ICE to adjust the contact probabilities $P(d)$. Other recent approaches use more elaborate strategies to call 'peaks' in Hi-C data. HiCCUPS, developed to detect sub-TAD chromatin interactions in a recent high-resolution Hi-C study by Rao et al.[11], compares each ICE-normalized interaction bin count in the contact matrix to the normalized counts of local neighborhoods; a peak is called only if the bin count is significant relative to all local comparisons and satisfies further filtering criteria. In another approach, Xu et al.[12] used a hidden Markov random field to explicitly model the spatial dependence of bin counts in the contact matrix, where the bin-level statistical model is a mixture of negative binomial distributions representing 'peak' and background states, and the expected counts for the background state are estimated by ICE or Fit-Hi-C.

Here we present an integrated model for detecting significant Hi-C interactions that systematically accounts for the dependence of interaction bin read counts on sources of bias like GC content and mappability, as in HiCNorm, as well as the dependence of random polymer ligation on genomic distance, as in Fit-Hi-C. Additionally, we explicitly model the zero-inflation and overdispersion of counts in the contact matrix by using a GLM approach based on hurdle regression[13]. By learning a null model that incorporates all these statistical properties of Hi-C contact matrix counts, our estimates of significance ($P$ values) reduce inflation in order to better identify direct interactions between regulatory or structural elements rather than nearby non-interacting loci. Our model can be estimated from a sampling of the data rather than working with the entire contact matrix. We focus our analysis on the Rao et al.[11] high-resolution in situ Hi-C data set in the GM12878 lymphoblastoid cell line and show that our method can identify significant interactions at the sub-TAD level, including DNA loops associated with CTCF and/or cohesin binding sites and enhancer-promoter interactions, as well as longer range (1.5–2 Mb) promoter–promoter loops. In particular, we identify a network of longer-range gene–gene interactions connecting the histone genes on human chromosome 6, potentially representing a specialized chromatin architecture at histone locus bodies. An implementation of our method, called HiC-DC (for 'Hi-C direct caller'), is available as an open source R package at https://bitbucket.org/leslielab/hic-dc (see also Supplementary Software 1).

## Results

**Hurdle negative binomial regression models Hi-C read biases.** We estimate a null or background model for Hi-C contact matrix counts using zero-truncated negative binomial (ZTNB) regression, also called hurdle regression[13]. By correctly modelling the zero inflation and overdispersion in the background model, we avoid inflating the significance of interaction bin read counts and reduce false positives. We assume that many of the contact matrix counts are well explained by the null model, which we use to estimate significance ($P$ values) of interaction bins with unusually high counts. For each interaction bin $(i,j)$, we take $y_{ij}$ to be the count of unique paired end reads joining intervals $I_i$ and $I_j$ and mapping within 500 bp of a restriction enzyme site on each side. We define a set of covariates $X_{ij}$ of the interaction bin, including the genomic distance $d_{ij}$ between intervals $I_i$ and $I_j$, and bias features like GC content and mappability of the effective sequence space for the pair of intervals[8], that is, the region within 500 bp of a restriction enzyme site (see Methods section).

The background model generates interaction bin read counts according to a two-step generative process: a Bernoulli distribution governs whether the count will be 0 (with probability $\pi_{ij} = \pi(X_{ij})$) or positive; if positive, then the read counts follow a negative binomial distribution with mean $\mu_{ij}$ and dispersion $\alpha$, where $\log \mu_{ij}$ is fit as a linear combination of B-spline functions (for the dependence on genomic distance) and bias-related covariates $X_{ij}$ (see Methods section). One rationale for the two-step generative process is that Hi-C libraries may not be complex enough to truly sample all random ligation events (and real interactions) that occur in the population of cells; rather, with some probability that depends on the covariates, ligated restriction fragments representing an interaction bin are not captured in the library, giving a zero count, similar to the 'drop-out' of lower expressed genes in single-cell RNA-seq[14].

For most analyses, we partitioned the genome into restriction fragments, concatenated 10 adjacent fragments, and used these genomic intervals to produce the Hi-C contact matrix.

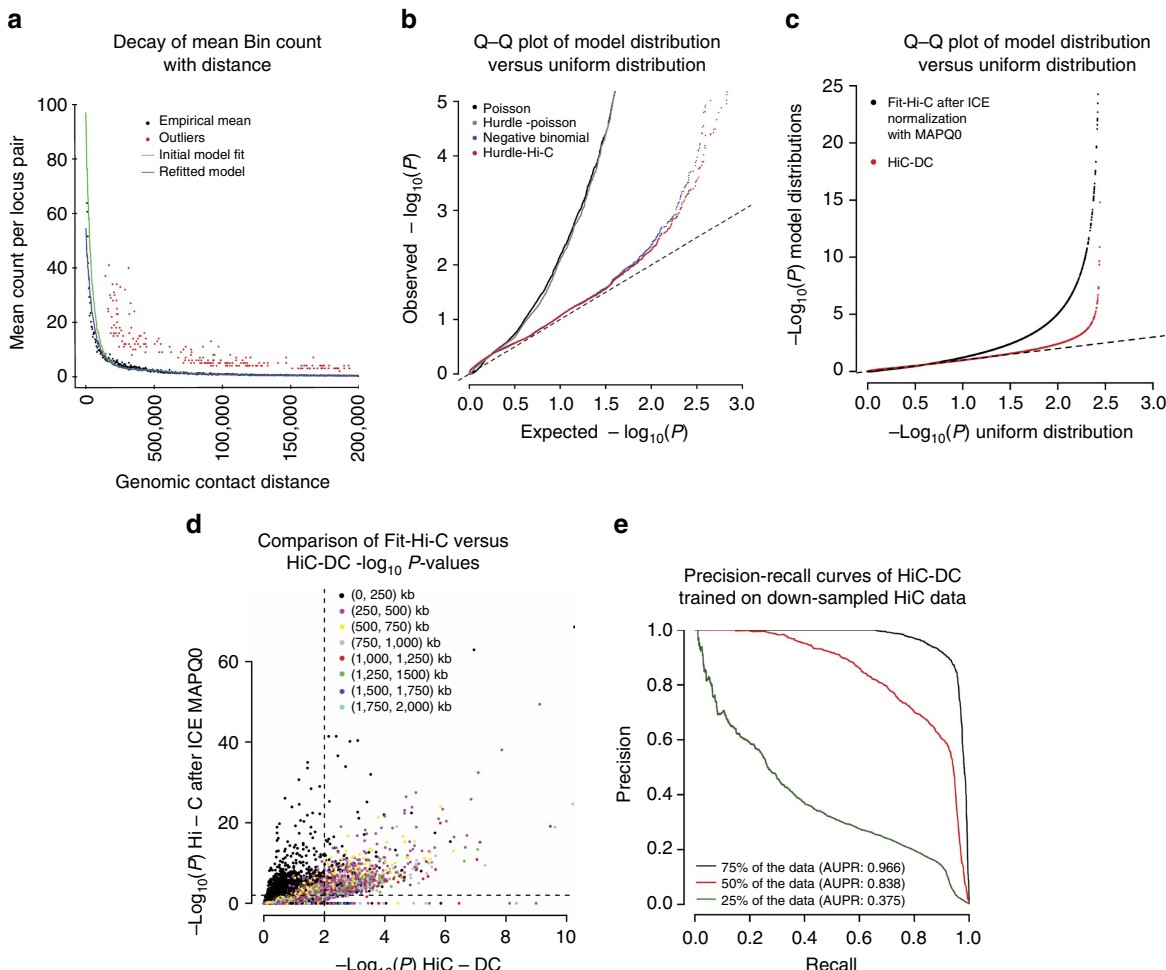

**Figure 1 | HiC-DC reduces inflation in estimates of statistical significance of Hi-C interactions.** (**a**) Expected interaction bin counts as a function of genomic distance according to the HiC-DC model for chromosome 1, using the Rao et al.[11] GM12878 Hi-C data set. (Green line) Initial model estimate, prior to outlier removal, plotting the mean of the hurdle regression model as a function of genomic distance and setting other covariates to their mean values. (Red points) Bin counts of outliers with mean counts in the top 3% of the zero-truncated negative binomial regression distribution as estimated by the model for the corresponding bins. (Blue line) Refit model, after removal of outliers. (Black points) Empirical mean bin counts for bins in genomic distance intervals. (**b**) Q–Q plots P values estimated by HiC-DC (zero-truncated negative binomial regression) as well as zero-truncated Poisson, negative binomial, and Poisson models (y axis) versus uniform distribution (x axis). All models were estimated on chromosome 1 of the Rao et al.[11] GM12878 data set. (**c**) Q–Q plots for HiC-DC and Fit-Hi-C, a method that uses ICE normalization to account for biases and estimates a binomial distribution using a spline fit. (**d**) Scatterplot of HiC-DC and Fit-Hi-C –log P values on chromosome 1 of the Rao et al.[11] data set for different genomic distance ranges. (**e**) Precision-recall curves for the detection of chromatin interactions by HiC-DC, defined as the interactions significant at FDR < 1% based on all read data, for different downsamplings of reads on chromosome 1 of the Rao et al.[11] GM12878 data set.

For high-resolution data from Rao et al.[11], this produced bins with median length ∼4 kb. For the medium resolution Hi-C data set for the human fibroblast cell line IMR90 from Dixon et al.[7], using a different restriction enzyme, the same procedure produced bins of median length ∼32 kb. An alternative procedure is to use uniform genomic intervals for the contact matrix and include the effective sequence space for pairs of intervals as an additional covariate[8]. We ran all models on interactions up to a genomic distance of 2 Mb to focus on sub-TAD structure.

We first analysed the high-resolution Rao GM12878 data set. Following Fit-Hi-C, we used two iterations of training to fit our null model. First, for each chromosome, we trained a ZTNB model using hurdle regression on a sample of 1% of all interactions bins $I_i$ x $I_j$ and confirmed that the dependence of the expected bin counts on $d_{ij}$ as estimated by the hurdle model indeed fit the empirical relationship (Fig. 1a for Chr 1; see

Supplementary Fig. 1 for other chromosomes). We then identified interaction bins with empirical counts $y_{ij}$ in the $P < 0.025$ tail according to the model, removed these bins from the training set as possible true interactions, and retrained the hurdle regression to obtain final parameters for the null model (Fig. 1a). We then used a quantile–quantile (Q–Q) plot to compare the P values produced by the ZTNB model to those drawn from the uniform distribution, which would be expected if all interactions came from the null model. Similarly, we plotted P values produced by applying (i) negative binomial regression, (ii) Poisson regression and (iii) Poisson hurdle regression to the same training data, again using two iterations of training. The Q–Q plots suggest that all alternative models inflated the significance of estimated P values (Fig. 1b for Chr 1; see Supplementary Fig. 2 for other chromosomes) and hence that modelling both the zero inflation and overdispersion is important for reducing false positive interactions.

We next ran both our model and the Fit-Hi-C algorithm on the same GM12878 data, using ICE normalization to adjust Fit-Hi-C $P$ values, as previously described; for both methods, we used uniform 5 kb bins and did not filter mappable reads for sequencing quality in order to get deeper coverage (corresponding to the 'MAPQG0' contact matrix from Rao *et al.*[11]). We removed diagonal ($d = 0$) interaction bins from HiC-DC since Fit-Hi-C filters out these bins. Comparison by Q–Q plots suggests that Fit-Hi-C produces $P$ values with inflated significance compared to HiC-DC (Fig. 1c, Chr 1; see Supplementary Fig. 3 for other chromosomes). This apparent $P$ value inflation of Fit-Hi-C relative to HiC-DC was more pronounced for the medium resolution IMR-90 Dixon *et al.*[7] data set (Supplementary Fig. 4). Genome-wide on the Rao[11] GM12878 data set, Fit-Hi-C used with ICE reported $\sim 793$ K interactions, while our approach reported $\sim 321$ K interactions, at a 1% false discovery rate (FDR) based on the Benjamini-Hochberg procedure. While there was some correlation between $-\log_{10} P$ values generated by the two methods within the same distance ranges, 74% of the interactions reported as significant based on the Fit-Hi-C binomial model would not be rejected by our null model (Fig. 1d, Supplementary Fig. 5, Supplementary Fig. 6). Using Fit-Hi-C without ICE normalization leads to a dramatic increase in the reported number of significant interactions, while filtering reads for quality (using the 'MAPQGE30' contact map) somewhat reduces Fit-Hi-C's interaction calls pre- or post-ICE; however, all versions of Fit-Hi-C call more non-diagonal interactions than HiC-DC at the same significance level (Supplementary Data 1, Supplementary Fig. 7).

We caution that there is no gold standard of true interactions against which false positive rates can be estimated, and Q–Q plot analysis presumes that most entries in the contact matrix are generated by the null distribution. However, since the binomial distribution used by Fit-Hi-C does not model overdispersion in read count data, similar to the Poisson regression variant of HiC-DC (Fig. 1b, Supplementary Fig. 2), we hypothesize that this feature of Fit-Hi-C may lead to $P$ value inflation.

To ask whether fine resolution interactions would be detectable at lower sequencing depth, we performed a downsampling analysis using 75, 50 and 25% of the reads (see Methods section). Using $P$ values based on 100% of the data ($\sim 129$ million paired-end reads) to define true interactions (FDR < 5%) and non-interactions (FDR > 10%), we asked if $P$ values estimated from downsampled reads could distinguish interactions from non-interactions. Precision-recall curves for this task (Fig. 1e) suggest that true interaction detection strongly degrades at the 25% sampling rate; at least 50% of the reads were required for reasonable (auPR of 84%) recovery of interactions at $\sim 4$ kb resolution.

We also confirmed the stability of the model relative to different 1% samples of training data. We trained HiC-DC on a fixed 1% sample of interaction bins (using uniform 5 kb bins) and defined the significant interactions at 1% FDR as our 'true interactions'. Then we retrained on 100 different 1% samples and computed, for each 'true interaction', the fraction of the models that detect the interaction (Supplementary Fig. 8). We found that 95% of 'true interactions' were detected by at least 90% of the models, and 90% of the interactions were detected by 100% of the models, showing stability of the training procedure.

**HiC-DC event calls are reproducible across experiments**. We also ran HiC-DC on additional data sets to examine the reproducibility of its event calls across biological replicates and different versions of the HiC-DC protocol, for example, the use of different restriction enzymes, as well as different versions of the

model, for example, uniform bins versus non-uniform bins. First, we considered the primary replicate and the largest secondary replicate Hi-C experiment in GM12878 from Rao *et al.*[11]. Since the secondary replicate has only slightly less coverage ($\sim 1.3 \times 10^9$ M aligned and filtered reads versus $\sim 1.5 \times 10^9$ M for the primary replicate), we used 5 kb uniform bins for each replicate and found good genome-wide concordance between $P$ values (Supplementary Fig. 9, $P = 0.52$). We also ran HiC-DC on two lower resolution Hi-C data sets[7] generated with different restriction enzymes, HindIII and NcoI, in mouse ES cells, using fixed bins at 50 kb resolution, and again found fair correlation between $P$ values (Supplementary Fig. 10, $P = 0.66$).

Although HiC-DC was not designed primarily as a Hi-C normalization procedure like the Yaffe and Tanay's method[6] or ICE[5], it can be used to normalize the contact matrix by dividing the observed bin count by the expected bin count estimated from the model ($O_{ij}/E_{ij}$). Following Yaffe and Tanay, we computed a heatmap of normalized counts ($O_{ij}/E_{ij}$) as a function of GC content in bin $i$ and bin $j$ for mouse ES Hi-C data generated with the *Hind*III and *Nco*I enzymes. Similar to previous findings[6], we observed preferential Hi-C contact patterns associated with low and high regional GC content, but these patterns were largely consistent between the two restriction enzymes (Supplementary Fig. 11).

Yaffe and Tanay also found a restriction fragment length bias in early Hi-C data sets[6], presumably due to differences in ligation efficiency. Analysing data from chromosome 1 of the Rao *et al.*[11] GM12878 data set with non-uniform binning, we found that observed counts for the 10% shortest bins are marginally higher than those of the 10% longest bins as a function of genomic distance, but this variation is small compared to the difference between the top and bottom 10% percentiles as estimated by the model (Supplementary Fig. 12). Therefore, the variation due to bin size is small compared to that of the modelled covariates. We further confirmed good concordance between $P$ values for the uniform (5 kb) and non-uniform (10RE) bin models (Supplementary Fig. 13, $P = 0.71$).

Finally, to provide a measure of experimental validation, we compared HiC-DC predictions (5 kb fixed bin model) for the Rao *et al.*[11] GM12878 data set against previously performed 3D-FISH validation experiments that confirmed four interactions between 30 kb intervals (L1, L2) along with negative controls (L2, L3). We took all contacts predicted by HiC-DC with endpoints overlapping (L1, L2) and (L2, L3) and examined both the maximal adjusted $P$ value and top 10 $P$ values among the contacts anchored in (L1, L2) and (L2, L3). HiC-DC correctly identified all of the significant events and did not assign significance to any of the contacts overlapping the control interaction bin (L2, L3) (Supplementary Table 1).

**HiC-DC enables detection of sub-TAD interactions**. To determine whether HiC-DC could reveal interactions involving individual regulatory or structural elements, we analysed the significant chromatin interactions called on Rao *et al.*[11] Hi-C data together with other epigenomic data sets for the B lymphoblastoid cell line GM12878 and chronic myelogenous leukemia cell line K562.

We first visualized the reported sub-TAD structure and our HiC-DC interactions at important B cell genes, together with DNase-sequencing (DNase-seq), RNA-sequencing (RNA-seq) and H3K27ac chromatin immunoprecipitation-sequencing (ChIP-seq) data, along with ChIP-seq data for CTCF and cohesin subunits SMC3 and Rad21, all generated by ENCODE in GM12878. For example, Fig. 2a shows the raw Hi-C interaction count matrix for a $\sim 700$ kb region encompassing the *BCL2* locus, with previously

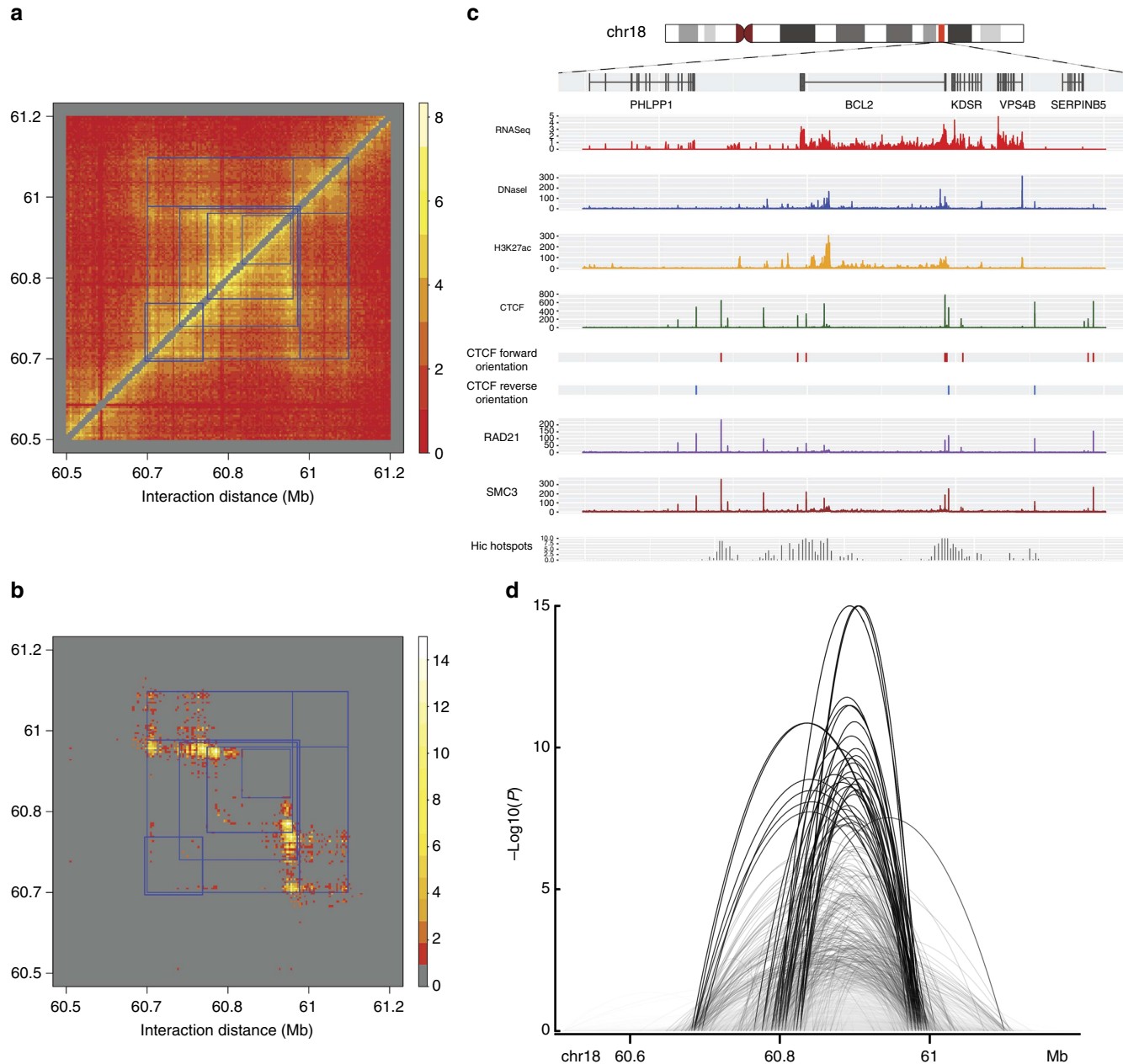

**Figure 2 | HiC-DC identifies identifies interactions associated with regulatory and structural elements at the sub-TAD level. (a)** Raw Hi-C count matrix from the Rao et al.[11] GM12878 data set for a ~700 kb region including the *BLC2* locus. Sub-TAD regions as called by Rao et al.[11] are shown as blue squares. **(b)** Significant interactions (–log10 P values) for the same region as estimated by HiC-DC. **(c)** Epigenomic tracks for GM12878 for the same region, showing DNase I hypersensitive sites, H3K27ac, ChIP-seq for components of the cohesin complex, and Hi-C hotspots as estimated by HiC-DC. **(d)** Sashimi plot depiction of significant interactions called by HiC-DC, showing chromatin looping between hotspots associated with regulatory and structural elements.

reported sub-TADs[11] drawn as blue boxes; Fig. 2b shows the interaction matrix with –log10 P values from HiC-DC in place of the raw counts. Consistent with the 'corner detection' strategy used by HiCCUPS[11], we see that three nested sub-TADs (~60.675 Mb–61 Mb) are indeed supported by significant 'corner' interactions—that is, interactions linking sub-TAD boundaries—while non-corner bins inside the sub-TAD have low significance. However, while a faint corner is detectable for smaller rightmost sub-TAD (~61 Mb–61.130 Mb), it is not supported by a significant HiC-DC interaction. Genome-wide, 'corner' interactions of sub-TADs reported by HiCCUPS were strongly enriched for significant interactions ($P < 10^{-16}$, KS test of –log10 P distribution of corners of true vs. randomized sub-TADs; Supplementary

Fig. 14), and HiC-DC P values accurately discriminated HiCCUPS interactions from a genomic-distance matched negative set of random interactions (Supplementary Fig. 15). By contrast, Fit-Hi-C tended to assign significant interactions to many loci within the sub-TAD, suggesting that sub-TAD 'corner' interactions—for example, interactions of pairs CTCF sites—are not well distinguished from nearby pairs of loci that may not represent direct interactions (Supplementary Fig. 16). HiCCUPS itself reports only a small number of interactions genome-wide, concentrated at genomic distances under 500 kb (~8K), so that while the majority of HiCCUPS predictions overlap with significant HiC-DC interactions computed at the same resolution (10 kb), HiC-DC also successfully operates at a longer genomic distance range

complementary to HiCCUPS (Supplementary Data 1, Supplementary Fig. 17, Supplementary Fig. 18).

To mechanistically interpret significant HiC-DC events, we examined other epigenomic signal tracks and the RefSeq genome annotation track alongside a 1D 'hotspot' summarization of HiC-DC analysis (Fig. 2c). The hotspot track shows, for each genomic interval, the maximum $-\log_{10} P$ value over interactions involving the interval; we restrict to interactions within the submatrix in Fig. 2b at FDR < 1% and display only interactions with distance > 50 kb for improved clarity. The arc plot representation in Fig. 2d shows these significant interactions as arcs joining genomic loci, where arc height represents statistical significance. The parallel epigenetic signal tracks show at least four distinct Hi-C hotspot regions: (i) a promoter proximal region that encompasses an intronic enhancer in the *BCL2* locus, the gene's DNase-accessible promoter that coincides with two CTCF peaks, and another CTCF peak upstream of the promoter (~61 Mb); (ii) a broad 3'-end hotspot region that includes an H3K27ac-marked intronic enhancer and multiple CTCF peaks (~60.8 Mb); (iii) a downstream hotspot that is co-occupied by CTCF and cohesin subunits SMC3 and Rad21 (~60.7 Mb); and (iv) an upstream hotspot with a strong CTCF peak as well as signal for SMC3 and Rad21. The arc plot (Fig. 2d) shows that the *BCL2* promoter hotspot has numerous significant contacts with both the 3'-end hotspot and the downstream hotspot, while the upstream hotspot also has a few significant contacts with the 3'-end hotspot and downstream hotspot. We do not see strong contacts between the 3'-end and downstream hotspots, even though both interact with the promoter. Consistent with previous observations[11], the interacting hotspots contained CTCF motifs with divergent orientation, while the non-interacting hotspots had motifs with the same orientation. We also examined the same locus in K562, using the Rao *et al.*[11] data set for this cell line (Supplementary Fig. 19). As expected, the B cell gene *BCL2* is not highly expressed in K562, and Hi-C interactions between the previously described hotspots are assigned low significance by our model. In particular, the interaction between the intronic enhancer and promoter of *BCL2* is lost in K562, and the enhancer there displays minimal DNase accessibility and H2K27ac.

We observed similar CTCF- and cohesin-mediated DNA looping insulating other important B cell genes. For example, the *IKZF1* locus is flanked by several upstream hotspots and several 3' end/downstream hotspots, all co-occupied by CTCF and cohesin to various degrees (Supplementary Fig. 20). Upstream and 3'-end/downstream elements interact with each other in pairwise fashion, insulating *IKZF1* from nearby genes. We also looked at the human β-globin locus to see if we observed interactions between the classic locus control region (LCR) and β-globin gene cluster[15] in the erythroleukemic cell line K562 (Supplementary Fig. 21). We indeed observed significant short-range interactions between the DNase accessible enhancers that make up the LCR, as well as short-range interactions involving the *HBB* and *HBD* loci. Furthermore, there is an interaction between the two CTCF sites flanking the region encompassing both the LCR and the gene cluster, albeit one of modest significance. Notably, these significant interactions are not observed at the locus in GM12878 (Supplementary Fig. 22), where the β-globin locus genes are not expressed.

**Distinct interaction types occur at different length scales.** HiC-DC analysis of the GM12878 data set provided sufficient resolution to annotate significant interactions by the epigenetic and genomic features of the interacting loci. We first asked whether significant interactions with specific epigenetic signals in GM12878—CTCF binding, DNase accessibility, and active

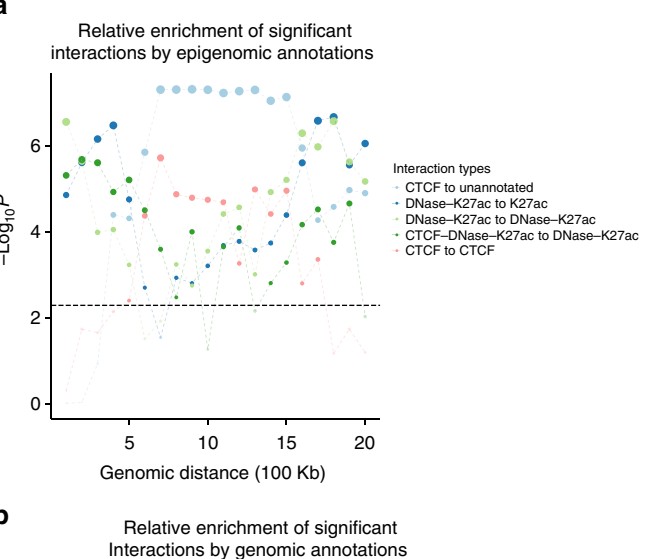

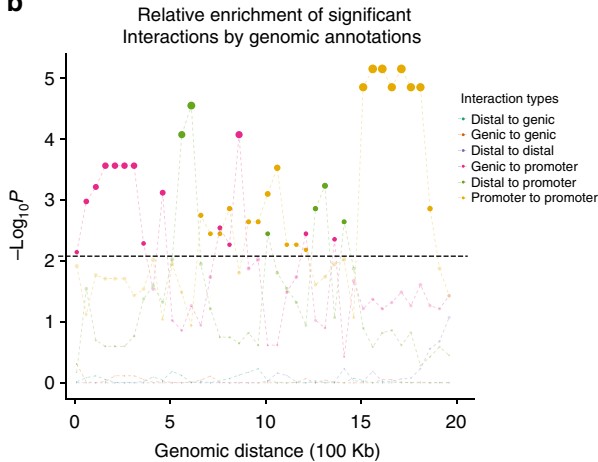

**Figure 3 | Distinct classes of structural and regulatory interactions occur at different distance ranges.** (**a**) Relative enrichments of significant HiC-DC interactions (FDR < 1%) for the Rao *et al.*[11] GM12878 data as annotated by epigenetic signals, as a function of genomic distance. For each 10 kb band, enrichment of interactions with each specific annotation were computed relative to the background prevalence of this annotation (by Fisher's exact test), and then the enrichment $P$ values for each annotation over 100 kb distance subranges were compared to all other annotations (Methods section). (**b**) Relative enrichments of significant HiC-DC interactions (FDR < 1%) for the Rao *et al.*[11] data as annotated by genomic location (promoter, gene body, or distal intergenic), as a function of genomic distance. Enrichment of interactions for each 10 kb band were computed as in **a**, then the enrichment $P$ values for each annotation over 50 kb distance subranges were compared as in **a**.

histone mark H3K27ac—were enriched at different length scales. We considered significant interactions (FDR < 1%) and determined, for each 10 kb distance band, enrichment of interactions with a specific annotation relative to background prevalence of the annotation (Methods section). We then compared the enrichment $P$ values for this annotation over 100 kb distance subranges to all other annotations (Methods section; see Supplementary Fig. 23 for absolute enrichment $P$ values). Relative to other annotations, CTCF-mediated interactions without other epigenetic signals (that is, CTCF-CTCF and CTCF-unannotated interactions) were most strongly enriched in the middle distance range of 700 kb–1.5 Mb (Fig. 3a), suggesting a preferred distance range for CTCF-associated structural loops (see also Supplementary Fig. 23). This length scale is consistent

with the reported length distribution of TADs (median length 880 kb), whose boundaries are enriched for CTCF binding sites. By contrast, interactions involving active regulatory elements, defined as DNase accessible and H3K27ac marked loci, were most enriched both at shorter distance ranges of <500 kb and, more surprisingly, at longer ranges of >1.5 Mb.

We then repeated the enrichment analysis by considering the genomic annotation—promoter, gene body, or distal intergenic— this time considering 50 kb distance subranges (Methods section; see Supplementary Fig. 24 for absolute enrichment *P* values). Here we found that, relative to other annotations, significant promoter-gene body interactions were most strongly enriched at the shorter distance range of <500 kb, consistent with intronic regulatory enhancers interacting with promoters (Fig. 3b). This analysis also revealed a strong enrichment of significant promoter–promoter interactions at longer distances of 1.5–2 Mb (Fig. 3b).

Genome annotation tools such as Segway[16] and ChromHMM[17] provide an automated way of assigning chromatin states to genomic intervals based on epigenomic data. We downloaded the seven-state combined Segway + ChromHMM segmentation for GM12878 from the UCSC Genome Browser and performed a similar enrichment analysis relative to these states, reporting absolute enrichments as a function of genomic distance rather than relative enrichments (Supplementary Figs 25 and 26). This analysis recapitulated some of our major findings, such as the enrichment of distal promoter–promoter interactions. We note that only 4% of genomic intervals previously annotated as CTCF binding sites based on ChIP-seq data were assigned a 'CTCF' state in this analysis—possibly because the CTCF states correspond to short segments and states with longer segments (for example, 'Repressed') dominate the annotations. Enrichments for CTCF-associated interactions were therefore less prominent in this analysis.

Previous megabase resolution Hi-C analyses described two chromatin compartments, called A/B compartments, defined by examining the sign of the first principle component of the contact matrix (or the correlation matrix generated from rows of the contact matrix). The A compartment is associated with open chromatin—that is, enriched for DNase hypersensitive sites and active histone marks—and the B compartment with closed chromatin. We binned the GM12878 Hi-C data at 100 kb resolution, determined A/B compartments as previously described (Methods section), and split significant interactions into three groups: within the A compartment, within the B compartment, and between A and B compartments. These groups displayed different patterns of enrichments for epigenomic signals and genomic annotations (Supplementary Fig. 27). Significant interactions within compartment A displayed greater enrichment for DNase-DNase and H3K27ac-H3K27ac marks at longer range genomic distances (>1.5 Mb) compared to those within compartment B (Supplementary Fig. 28). Longer-range promoter–promoter interactions were also uniquely enriched within compartment A. Significant interactions between compartment A and compartment B displayed little enrichment for epigenomic signals or genome annotations at any distance range (Supplementary Fig. 27).

Similarly, we segregated HiC-DC predicted interactions into inter-TAD and intra-TAD contacts based on previous TAD annotations[7] and found different patterns of enrichments for epigenetic signals and genomic annotations. For example, significant inter-TAD interactions showed greater enrichment for DNase-DNase and K27ac-K27ac marks at longer range genomic distances (>1.5 Mb) compared to significant intra-TAD interactions (Supplementary Fig. 29). Longer-range promoter–promoter interactions were uniquely enriched within inter-TAD regions (Supplementary Fig. 30).

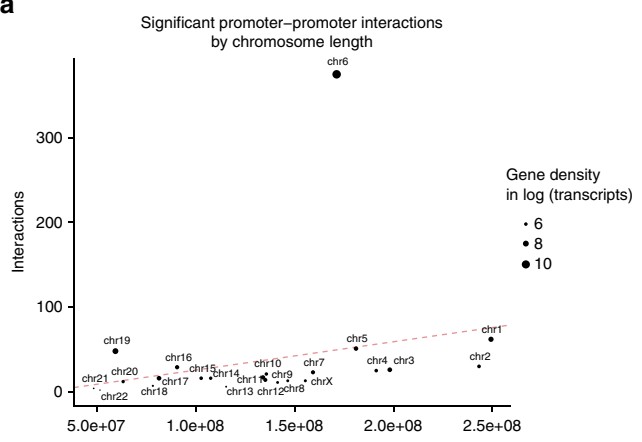

a

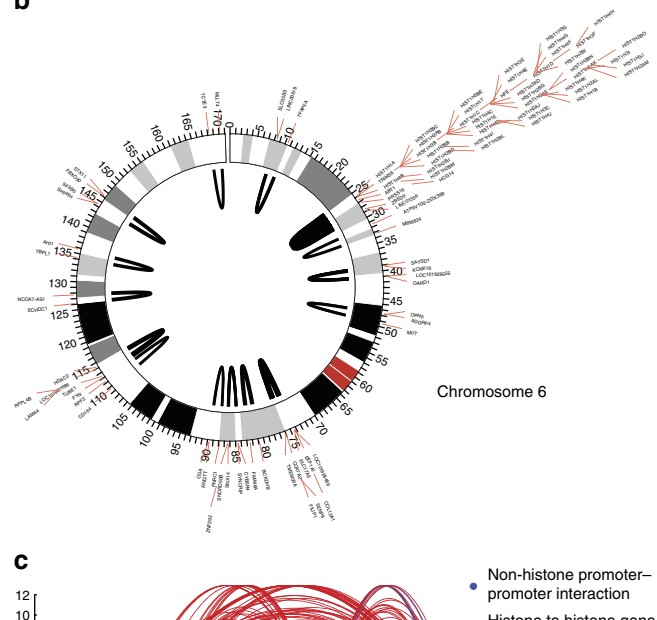

b

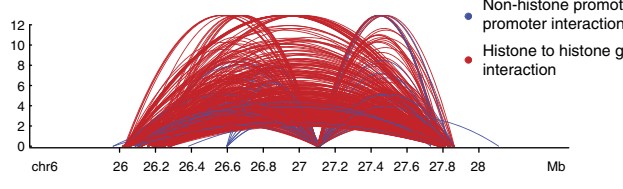

c

**Figure 4 | Long-range promoter–promoter interactions are enriched at the histone locus on chromosome 6.** (**a**) Number of significant long-range (1.5—2 Mb) promoter–promoter interactions per chromosome plotted versus chromosome length, showing striking enrichment of such interactions on chromosome 6. (**b**) Circos plot of long-range (1.5–2 Mb) promoter–promoter interactions on chromosome 6. Opacity of arcs indicates density of Hi-C contacts, showing a cluster of interactions at the histone locus. (**c**) Significant promoter–promoter interactions linking histone gene pairs (red arcs) or non-histone gene pairs (blue arcs) at the histone locus on chromosome 6.

**HiC-DC identifies long-range histone gene interactions.** To investigate the long-range promoter–promoter interactions from our enrichment analysis, we examined the distribution of these interactions by chromosome. Strikingly, we found that significant 1.5–2 Mb promoter–promoter interactions were highly concentrated on chromosome 6, both in absolute number (Supplementary Fig. 31) and when normalized by chromosome length (Fig. 4a). Of the 90 genes on chromosome 6 involved in these interactions, over 40% (37/90) of them were

replication-dependent histone genes, all but one of them joined by a dense connected interaction network; other genes participated in small clusters of interactions of 8 or fewer genes (Fig. 4b, Supplementary Fig. 32). Interactions between histone promoters were also recently reported in an analysis of capture Hi-C data[18]. A visualization of the significant promoter–promoter interactions in the 26–28 Mb region of chromosome 6 shows three subclusters of histone genes that appear to interact with each other in long-range chromosomal looping (Fig. 4c). Histone genes are short (median length $\sim 410$ bp), and multiple histone genes can fall into a single genomic interval in our analysis. Globally, long-range 'promoter-promoter' interactions tend to arise from gene-dense regions and involve genes with short length (Supplementary Fig. 33) and therefore are more accurately described as 'gene-gene' interactions.

Expression of replication-dependent histone genes is restricted to S phase, when massive production of histone proteins is required to package newly replicated DNA[15]. Histone genes also require specialized mRNA processing, since they are not polyadenylated but rather contain a $3'$ end hairpin structure. Factors required for hairpin recognition and $3'$ cleavage of histone genes, including hairpin binding protein (HBP, also called SLBP) and the U7snRNP, are found in high concentrations in nuclear bodies called histone locus bodies[19], which have been most extensively studied in *Drosophila*. It is possible that the network of histone interactions we observe is the result of chromosome looping in S phase to bring all histone genes on chromosome 6 into close proximity to facilitate mRNA processing at the histone locus body. A recent single cell study in GM12878 in fact estimated that 40% of cells are in S phase in cell culture conditions[20].

## Discussion

We have presented a principled statistical approach for detecting significant chromatin interactions from Hi-C read count data. We account for systematic sources of bias, including distance-dependent random polymer ligation as well as GC content and mappability, in a single model rather than using an *ad hoc* normalization scheme prior to analysis. HiC-DC allows us to detect significant sub-TAD interactions, interpret multiple interacting Hi-C hotspots at developmentally important genes, and identify regions with specialized chromatin organization, such as the network of long-range interactions between histone genes on chromosome 6.

Our model assumes that most interaction bin counts are well explained by the null distribution. Q–Q plot analysis suggests that methods that fail to model overdispersion of read count data—including Poisson regression variants of HiC-DC as well as Fit-Hi-C—may be prone to inflation of $P$ values. However, in the absence of either a 'gold standard' of true interactions or an experimental procedure to empirically produce a Hi-C null distribution, we cannot make definitive claims about accuracy. There are several directions for improvement of the HiC-DC null distribution, which currently uses a fixed dispersion estimate per chromosome and does not account for hierarchical structure in the data (non-independence between bin counts). Training HiC-DC on 25 kb slices of the genomic distance distribution suggests that the dispersion parameter does vary with distance (Supplementary Figs 34 and 35) and is close to 0 for interactions bins of $>1$ Mb. A future extension could develop an appropriate parametric form for $\alpha(d)$ within the GLM framework. Moreover, two loci separated by genomic distance $d$ that lie inside a TAD likely are closer in 3D and generate higher read counts than loci at distance $d$ in adjacent TADs; however, all interaction bins at distance $d$ are treated equivalently by the model. While this

assumption promotes detection of the most significant looping interactions that support the sub-TAD structure, rather than other pairs of loci within the loops that are brought closer by these interactions, HiC-DC sensitivity might be improved by modelling the hierarchical structure among interaction bins. A further extension would be a simplified version of HiC-DC for inter-chromosomal interactions: for each pair of chromosomes, the matrix of bin counts between pairs of intervals would be output values in the regression model, with the bias covariates as before but no genomic distance dependence.

HiC-DC incorporates covariates such as GC content and mappability directly in the regression model rather than trying to first rebalance the count matrix using an approach like ICE. Our method is both more direct and more scalable: when applied to high-resolution Hi-C experiments with large interaction matrices, matrix rebalancing algorithms can become numerically unstable and lead to unpredictable results. Indeed, to apply ICE to the large interaction matrices in their study, Rao *et al.*[11] masked a large number of interaction bins (more than 20% on chromosomes 9, 13, 14, 15, 21, 22)—including centromeric, telomeric, and other low-count regions—with 'NaNs' in order to ensure ICE convergence (Supplementary Fig. 36). Our approach requires no such masking while appropriately dealing with zero counts and low mappability bins.

Analysis of Hi-C data may also be confounded by phases of the cell cycle. We observed a dense network of long-range (1.5–2 Mb) histone gene interactions on chromosome 6, consistent with their co-localization in the histone locus body to enable efficient mRNA processing in S phase. However, it is also possible that this chromatin organization is maintained outside of S phase to poise the histone genes for rapid mRNA expression once the appropriate transcription and RNA processing factors are available. Future Hi-C studies in synchronized cells will resolve these questions.

## Methods

**Data preprocessing.** Hi-C paired end (PE) reads were mapped to the hg19 reference genome using the Burrows-Wheeler Aligner (BWA). The paired ends were mapped independently and matched later in the analysis, with sequence alignments performed in parallel on a cluster. We used the BWA single end aligner *bwa-sw* but found that BWA *bwa mem* worked equally well. The short read version of aligner, *bwa-aln*, did not perform well for aligning longer length paired end reads for Hi-C data.

As a filtering step, we only used PE reads where each end uniquely mapped to the reference genome, and clonal reads from PCR amplification were collapsed. In addition, PE reads that did not map within 500 bp from a restriction enzyme (RE) site were eliminated.

**Calculating local genomic features.** We divided the genome into consecutive disjoint intervals, with RE sites as breakpoints, and merged every ten contiguous RE fragments to produce larger non-overlapping intervals. We used the GM12878 in situ Hi-C dataset by from Rao *et al.*[11]. In this Hi-C experiment, a 4 base cutter (MboI) was used for chromatin fragmentation and yielded intervals (after merging every 10 RE fragments) with a median width of $\sim 4$ kb. Each pair of intervals defines an interaction bin for counting PE reads, and we developed a model for assessing the significance of these counts based on local genomic features and the linear distance between the interval midpoints.

For each interval, we computed the average GC content and mappability over regions within 500 bp of a RE site. For each interaction bin, we standardized the log transformed GC content and mappability features to obtain GC content and mappability covariates for the GLM described below.

The HiC-DC model can also be used with a uniform binning of the genome. In this case, an additional covariate called the effective sequence space is included in the model[8]. For each interaction bin, we compute the fraction of each of the corresponding genomic intervals that is within 500 bp of a RE within the interval; the effective sequence space is the product of these fractions.

**Modelling interaction bin count data.** The read counts for each pair of intervals ('interaction bin') from high-throughput conformation capture are overdispersed relative to the Poisson distribution (Fig. 1b). The negative binomial (NB) distribution is typically used for modelling overdispersed Poisson read count data.

However, the NB distribution cannot handle an inflation of zero observations (zero count bins), which is typical in Hi-C data, perhaps due to a 'drop-out' phenomenon where less frequent interactions are never captured in the sequencing library, as in single-cell RNA-seq[14]. One option is to use a zero-truncated negative binomial (ZTNB) regression model. Under this distribution, a Bernoulli model governs the binary outcome of whether a count variable has a zero or positive value. If the outcome is positive, the 'hurdle' is crossed, and the conditional distribution of the non-zero counts is governed by a negative binomial distribution. This model is sometimes called a hurdle negative binomial model. We used a GLM based on zero-truncated negative binomial regression to model the Hi-C read counts, and we used the fitted model to estimate the statistical significance (P value) to the Hi-C interaction bin counts.

Let $Y = \{y_{ij}\}$ represent the Hi-C contact map of intra-chromosomal interactions, where $i$ and $j$ are a pair of genomic intervals (as described above) and the tuple $(i,j)$ defines an interaction bin. Each bin has an associated vector of covariates, which we denote as $\mathbf{X} = \{x_{ij}^{\text{dist}}, x_{ij}^{gc}, x_{ij}^{\text{map}}\}$. $y_{ij}$ is a random variable that follows a zero-truncated negative binomial distribution. The regression model is defined as

$$P(Y = y_{ij}|\mathbf{X}) = \begin{cases} \pi_{ij}, & y_{ij} = 0 \\ \dfrac{(1-\pi_{ij})}{\left(1-\left(1+\alpha\mu_{ij}\right)^{\alpha^{-1}}\right)} f(y_{ij}; \mu_{ij}, \alpha), & y_{ij} > 0, \end{cases}$$

where the distribution $f(y_{ij}; \mu_{ij}, \alpha)$ is a negative binomial distribution with dispersion parameter $\alpha$. The negative binomial mean parameter $\mu_{ij}$ is described with a log-linear model

$$\log\left(\mu_{ij}\right) = \beta_o + \sum_k \beta_k B_{k,l}(x_{ij}^{\text{dist}}) + \beta_{gc} x_{ij}^{gc} + \beta_{\text{map}} x_{ij}^{\text{map}},$$

where $\beta_o$ is the intercept term, and $\beta_{gc}$ and $\beta_{map}$ are coefficients for GC content and mappability features, respectively. We modelled the relationship between genomic distance and contact significance (Supplementary Fig. 1) with a third order B-spline ($l = 3$), which takes the distance covariate as input, and the $\beta_k$ correspond to the coefficients of B-spline basis functions:

$$B_{i,0}(x) = \begin{cases} 1, & t_i \leq x < t_{i+1} \\ 0, & x \geq t_{i+1}, x < t_i \end{cases}$$

$$B_{i,j+1}(x) = \alpha_{i,j+1}(x)B_{i,j}(x) + [1 - \alpha_{i+1,j+1}(x)]B_{i+1,j}(x)$$

where

$$\alpha_{i,j}(x) = \begin{cases} \frac{x - t_i}{t_{i+j} - t_i}, & t_{i+j} \neq t_i \\ 0, & t_{i+j} = t_i \end{cases}$$

and $t_i$ are ordered knots. The number of inner knots equals the degrees of freedom minus the order of the B-spline. We use six of degrees of freedom for fitting the spline and hence three inner knots. The inner knots are 25, 50 and 75% quantiles of the linear genomic distance, and the boundary knots are 0 and 2 Mb.

To increase the statistical power of the model, we removed bin counts that exceed the 97.5% percentile of null distribution that we deem as positive outliers, corresponding to potentially non-random contacts, and then refit the model to the remainder of the data. We use the function hurdle in the R library pscl to fit the zero-truncated negative binomial regression model.

We assume that the majority of interaction bins are generated by the null distribution, that is, do not represent significant interactions. HiC-DC models the null read count distribution as described above, and it assesses the significance of unexpectedly large interaction bin counts based on the corresponding estimated ZTNB distributions. To do this, after model fitting, each bin within a suitable distance range is assigned a P value by subtracting from 1 the sum of probabilities for all values less than the observed read count for the bin. Significant bins can then be selected with adjusted P values controlling for FDR based on the Benjamini-Hochberg procedure.

**Annotating bins with genomic and epigenomic labels.** To add epigenomic annotations to our bins, we first performed IDR on two replicates of DNaseI ChIP-seq peaks (ENCFF001WFR and ENCFF001WFS by ENCODE), as well as two replicates for CTCF ChIP-seq peaks (ENCFF001XPJ and ENCFF001XPK by ENCODE) to obtain a set of reproducible peaks for each data set. Only one replicate for H3K27ac was available (ENCFF001SUG), which precluded IDR and so we took those peaks as they were. We merged all three types of peaks together into one catalogue of peaks by successively merging the GRanges objects. Where peaks of different types overlapped, we concatenated the peak labels.

We then annotated each of the genomic bins using the catalogue of peaks, by transforming our bins into a GRanges object, and assigning peak annotations to bins where they shared overlap as determined by the findOverlaps function in GRanges. Each contact inherited the annotations of both its genomic bins.

To assign genomic annotations to the bins, we downloaded an annotation of all known genes from RefSeq (built against GRCh38), and split these into known genes and known transcripts. We used the transcripts along with the following rules to annotate each bin. If the bin was within 2kb of a transcription start site, it was labelled as containing a promoter. If the bin overlapped with a known exon, it was labelled as exonic. If the bin overlapped with an intron, it was labelled intronic.

Finally, if none of the previous were satisfied, it was labelled as distal intergenic. We later merged intronic and exonic labels into gene-body (or genic), since exonic bins were so sparsely distributed.

**Calculating annotation enrichment among significant contacts.** To ask whether significant interactions with specific epigenetic signals in GM12878 were enriched over different distances, we partitioned all contacts by distance (excluding self-ligating contacts) in increments of 10 kb, covering distances from up to 2 Mb. For each 10 kb sized element of the distance partition, that we termed a distance band, and each epigenomic label, we collected all contacts that fell within this distance band into significant (FDR < 1%) contacts bearing the label, significant contacts not bearing the label, not-significant contacts sharing the given label, and not-significant contacts not sharing the label. We used a Fisher's exact test on the contingency table of those four groups to determine the enrichment of each label among significant contacts in that distance band (results depicted in Supplementary Fig. 23).

To examine the enrichment signal across longer distances, we pooled the results from the Fisher's exact test described above into contiguous regions of 100 kb. This gave us 10 observed levels of significance for each annotation over each 100 kb region. We applied a Wilcoxon rank-sum test on these P values to get a relative enrichment value for each annotation over the longer intervals, which we display in Fig. 3a.

For the genomic annotations, we repeated the same methodology of testing for enrichment in 10 kb distance bands for bins annotated as either a promoter, gene body, or distal intergenic. We pooled the results over 50 kb contiguous regions, and applied a Wilcoxon rank-sum test on the P values to get a relative enrichment value, which we display in Fig. 3b.

**Genomic compartment analysis.** To identify A and B compartments in the Rao et al.[11] GM12878 data set, we merged adjacent interaction bins, combining their counts to form meta-bins with 100 kb resolution. We computed a correlation matrix from this coarse-grained contact matrix, and computed its first principal component. Following the procedure in [1], the sign of the first principal component for each meta-bin was used to define the compartment label (A or B). We found that the A compartment meta-bins were enriched for DNase hypersensitive sites, and so labelled the A compartment meta-bins as 'open', then the B compartment meta-bins 'closed'.

For the compartment enrichment analysis, we applied the label of each meta-bin to its constituent bins. We then partitioned all contacts into three groups: contacts with both endpoints in open regions, contacts with both endpoints in closed regions, and contacts with one open and one closed endpoint. For each of the three sets, we repeated the enrichment analysis as in Supplementary Fig. 14.

**Downsampling analysis.** For the downsampling analysis, we performed training on chromosome 1 in Rao et al.[11] GM12878. We downsampled the number of reads from the Hi-C interaction matrix by taking the counts for each element (that is, each pair of genomic loci) and transforming them into a list of paired end reads of size equal to the counts. From this list of all reads, we subsampled randomly without replacement and reassigned each of the sampled reads to their corresponding elements in the new interaction matrix. We trained HiC-DC on samples in proportions of 75, 50 and 25% from the original contact matrix. We used HiC-DC to calculate FDR-adjusted $-\log_{10} P$ values for each interaction in each of the three sub-sampled contact matrices. These P values were used to predict the significant interactions from the full interaction matrix. The FDR-adjusted P values for the full Hi-C contact were used to define our ground truth labels. Interactions with adjusted P values < 0.05 were labelled as positive, and interactions with adjusted P values > 0.1 were labelled as negative. We excluded all bins with adjusted P values between 0.05 and 0.1 from the analysis. For each of the down-sampled contact maps, we plotted precision-recall curves as shown in Fig. 1e.

**Data availability.** Hi-C data sets analysed in this study were obtained through Gene Expression Omnibus accession codes GSE63525 (GSM1551550—GSM1551578, GSM1551618—GSM1551623) and GSE35156 (GSM862720—GSM862723, GSM892307). Source code and documentation for HiC-DC is available as an R package through a git repository located at https://bitbucket.org/leslielab/hic.dc.

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

## Acknowledgements

This work was supported in part by NIH/NHGRI grants HG006798 and HG007893 to C.S.L. We would like to thank Iestyn Whitehouse for helpful discussions and Ferhat Ay for assistance with the Fit-Hi-C software.

## Author contributions

M.C. developed and implemented the statistical model, performed statistical analyses including method comparisons and detailed views of DNA looping at specific loci, and contributed to writing the manuscript. L.Z. performed statistical analyses including genome-wide enrichment analyses and mapping of histone locus interactions and contributed to writing the manuscript. M.S. contributed to optimizations of the HiC-DC code and statistical analyses. A.G. processed the Hi-C data to produce count matrices of interaction bins, parsed epigenomic datasets to annotate the bins, and advised on the software implementation. R.P. advised on algorithm development. O.E. helped to supervise the research. C.S.L. helped to develop the statistical model, supervised the research, and wrote the manuscript.

## Additional information

**Competing interests:** The authors declare no competing financial interests.

