## [Peer Review File · Nature Communications]

Reviewers' Comments:

Reviewer #2 (Remarks to the Author)

In this paper, the authors proposed HiC-DC, a zero-truncated negative binomial (ZTNB) regression based approach, to normalize Hi-C count data and detect significant chromatin interactions. This novel method combines two critical Hi-C data analysis tasks, normalization and peak calling, into an integrated framework. The authors did comprehensive data analysis, and several novel biological findings are very interesting. HiC-DC addresses an important research question, and has the potential to become a useful tool for the general research community. The paper is also well written. I have a few comments which may further improve the current work. Here are my specific comments:

Major comments:

1. HiC-DC used a global background model to re-analyze and call peak the GM12878 data in Rao et al 2014 cell paper. However, Rao et al paper used a local background model HiCCUPS to identify chromatin loops. Based on different background models, the resulting peaks or loops are different. The authors can compare HiC-DC and HiCCUPS, and give some guidance of peak calling for general audience.
2. For the different peak calling results among HiC-DC, Fit-Hi-C and HiCCUPS, it would be more convincing if the authors can design some FISH experiment as validation.
3. What's the reproducibility of HiC-DC? If we apply HiC-DC to Hi-C data with biological replicates, can we get similar peak calling results? Similarly, if we apply HiC-DC to Hi-C data with different enzymes (HindIII vs. NcoI vs. Mbol) or different protocols (in situ Hi-C vs. dilute Hi-C), can we get reproducible peak calling results?
4. Page 10, line 252: the authors found distinct classes of interactions are enriched at different length scales. I am curious whether such finding is affected by TAD structure. It is known that intra-TAD interactions are much higher than inter-TAD interactions. The authors can divide all interactions into two groups: intra-TAD and inter-TAD interactions, and check whether the same conclusion holds within both groups.
5. Page 13 line 350. The authors mentioned the advantage of HiC-DC over the popular normalization method ICE. I would love to see more in-depth analysis along that direction. For example, can HiC-DC achieve higher reproducibility of normalized Hi-C data between biological replicates than ICE? Can HiC-DC achieve better effect of bias removal than ICE? One possible analysis is to check whether normalized Hi-C data, by HiC-DC or ICE, is still affected by local genomic biases. Heatmaps similar to Yaffe and Tanay 2011 Figure 1d, 1f and 1h will be very helpful.
6. Page 15 line 411. The authors mentioned that when applying HiC-DC with a uniform binning of the genome, they add an additional covariate of effective sequence space. I wonder whether such effective sequence space is also useful in non-uniform binning? When merging every 10 RE fragments into interval, different intervals can still have different effective sequence spaces. In addition, according to Yaffe and Tanay 2011 Figure 1c and 1d, the unbalance fragment size leads to ligation efficiency bias. The author may also explore such fragment bias in HiC-DC with non-uniform binned Hi-C data.
7. Page 16, line 437: The authors used dispersion parameter α to account for over-dispersion in negative binomial distance. Since Hi-C data show high heterogeneity, does the constant α

fit data well? Can they allow α to be a function of 1D genomic distance? Or different dispersion parameters for intra-TAD and inter-TAD interactions?

8. Figure S3 and S4: It is interesting to see Fit-Hi-C provides P-value with inflated significance compared to HiC-DC. Is it because the zero-truncated negative binomial model used in HiC-DC fits data better than binomial model used in Fit-Hi-C? The authors may provide some numerical evidence of model fitting.

Minor comments:

1. I cannot find Figure 1. In the combined pdf file, page 24 is blank. The authors need to add Figure 1 in the revision.

2. The authors can add some description of computational details of HiC-DC. What's the memory requirement and time cost of using HiC-DC to conduct genome-wide peak calling on the high resolution Hi-C dataset?

3. Page 6, line 138. HiC-DC trains the ZTNB model using a sample of 1% of all interactions. The authors may provide more details on how to sample this 1%? Is ZTNB model and HiC-DC result sensitive to different 1% samples? Some sensitivity analysis and robust test will be more convincing.

4. Can we apply HiC-DC to normalize inter-chromosomal interactions and identify inter-chromosomal interaction peaks?

5. In Figure S10 and S11, it is better to also show raw Hi-C contact matrix and HiC-DC results, similar to Figure S9a, S9b.

Reviewer #3 (Remarks to the Author)

In this work, Carty et al. developed an innovative statistical model (HiC-DC) to detect significant interactions in Hi-C data. This model is based on a hurdle negative binomial regression that takes consideration of many known biases in Hi-C interaction matrices, such as GC content, mappability, and the distance dependence effect due to random polymer collision. The innovation of this proposed method is that the authors addressed the zero-inflation and overdispersion, which are commonly seen in Hi-C data, but have not been effectively addressed in other existing Hi-C analysis software.

Overall, the manuscript is well written and the analysis was properly done. The figures are clear and precise. The software performed well and their application to high-resolution Hi-C data also lead to some interesting discoveries regarding gene regulatory activities, such as different structural and regulatory interactions are enriched in distinct distance ranges. The work will provide a great tool for the 3D genome organization community and novel biological findings as well.

That being said, I do have some comments and suggestions that I hope the authors can address:

1. Overall, the authors did a great job reviewing the previous efforts and the remaining challenges in analyzing and modeling Hi-C data. I do suggest the authors reviews a couple of other Hi-C peaking calling methods, namely, HiCCUPs by Rao SS et al 2014, and Zheng Xu 2015 et al.

2. Line 158: the authors' claim that there is P-value inflation of by Fit-Hi-C need to be further justified. More details of how this figure was generated and more discussion will be helpful.

3. The authors continued to argue that HiC-DC reported much less significant interactions than Fit-Hi-C (1 million vs 3 million). Can the authors draw a Venn Diagram and show where the differences are? Further, for the differential peaks, can they compare with other types of data (say ChIA-PET to get an evaluation)?

4. It is interesting that the authors discovered different classes of interaction are enriched in different length scales. For example, the CTCF-mediated interactions are most enriched in the distance range 700Kb -1.5Mb. However, the underlying biological implication of these differences are not fully discussed.

5. Figure 3. What is the rationale to plot "DNase-K27ac to K27ac", "CTCF-DNase-K27ac to DNase-K27ac"? Since GM12878 is an ENCODE cell line, why not just use their definition of enhancers, promoters etc.

6. Figure 1a, the labels for the blue line and green line are inconsistent with description in its figure legend.

Response to Reviews

Reviewer #1 comments:

In this paper, the authors proposed HiC-DC, a zero-truncated negative binomial (ZTNB) regression based approach, to normalize Hi-C count data and detect significant chromatin interactions. This novel method combines two critical Hi-C data analysis tasks, normalization and peak calling, into an integrated framework. The authors did comprehensive data analysis, and several novel biological findings are very interesting. HiC-DC addresses an important research question, and has the potential to become a useful tool for the general research community. The paper is also well written. I have a few comments, which may further improve the current work. Here are my specific comments:

Major comments:

1. HiC-DC used a global background model to re-analyze and call peak the GM12878 data in Rao et al 2014 Cell paper. However, Rao et al paper used a local background model HiCCUPS to identify chromatin loops. Based on different background models, the resulting peaks or loops are different. The authors can compare HiC-DC and HiCCUPS, and give some guidance of peak calling for general audience.

HiCCUPS is an event-calling algorithm that compares the (normalized) count of each element in a Hi-C contact matrix to a series of averaged (normalized) counts from elements in different local neighborhoods. For a given contact with genomic bin endpoints $[i, j]$, the local neighborhoods are defined by considering all bins in $[i \pm d, j]$ (the horizontal neighborhood), $[i, j \pm d]$ (the vertical neighborhood), $[i-d, j-d]$ (the bottom left neighborhood) and $[i \pm d, j \pm d]$ (the donut neighborhood) for different values of d . The significance of each contact is assessed in the context these neighborhoods by testing the null hypothesis that the observed read count for that contact was generated by a Poisson random variable with parameter λ , with λ determined by a neighborhood average. A contact is considered a significant peak only if it is considered significant in all of its neighborhoods. There is further filtering of significant peaks based on criteria of fold-enrichment, collapsing proximal enriched contacts into single peaks, and other *post hoc* adjustments.

Broad differences exist between the HiCCUPS and HiC-DC models. HiC-DC operates directly on a matrix of counts, using bin-associated features to correct for biases, whereas HiCCUPS requires preprocessing by ICE to produce a normalized contact matrix. HiC-DC uses a single model to describe Hi-C contact counts, while HiCCUPS uses a collection of local models. Finally, HiCCUPS leverages data binned at multiple resolutions for event calling, while HiC-DC considers only data at one resolution. In practice, the computational resources required to run HiCCUPS can greatly exceed those for HiC-DC. For Hi-C data sets with high resolution (e.g. the primary GM12878 replicate of Rao et al.), HiCCUPS requires substantial computation time both for ICE normalization (which also suffers from numerical instability, as we describe in the manuscript), as well as specialized GPU hardware for assigning significance to peaks. In contrast, HiC-DC runs on common multi-core CPUs and processes the same high-resolution data set in several hours. Rao and colleagues had to remove large numbers of interaction bins in order for the ICE normalization procedure to converge (personal communication).

Moreover, HiCCUPS only considers interaction bins with a linear genomic distance greater than 50kb at 10kb resolution to make sure there is sufficient local neighborhood for any arbitrary bin. Overall, HiCCUPS detects a smaller number of significantly enriched peaks compared to other methods (see **Supp. Fig. 17** and **Supp. Table 1**), reporting only ~8K interactions genome-wide. In particular, HiCCUPS reports few interactions (352 out of ~8K) at genomic distance greater than 1Mb, although CTCF-associated chromatin loops are enriched at this range (**Supp. Fig. 18**).

We also included an analysis to see how well HiC-DC could detect sub-TAD interaction produced by the HiCCUPS algorithm on GM12878 data from Rao et al. 2014. We considered interactions corresponding

to sub-TADs as annotated by Rao et al. as positive examples and took negative examples to be random genomic regions with the same length genomic distributions as the annotated sub-TADs. For each sub-TAD, we used as our prediction value the maximum $-\log_{10} P$ value of all Hi-C interaction bins overlapping the “corner” of the sub-TAD (i.e. the HiCCUPS peak call). For this task, HiC-DC recovered Rao et al. sub-TADs with a strong auPR (area under the precision-recall curve) of 0.89 (**Supp. Fig. 15**).

2. For the different peak calling results among HiC-DC, Fit-Hi-C and HiCCUPS, it would be more convincing if the authors can design some FISH experiments as validation.

We agree in principle with the reviewer’s comment that supplemental FISH experiments would help validate the peaks called by HiC-DC. However, as a computational lab, we are not in the position to easily carry out new FISH experiments. To try to address the request, we instead examined the four chosen loci from the FISH experiments provided by Rao et al. (cf. Table S5 in Rao et al. 2014). Briefly, they performed 3D-FISH on four peaks to confirm that the 30kb bin loci (L1, L2) anchoring their predicted events were closer in 3D than all contacts between L2 and a control locus L3.

We took all contacts identified by HiC-DC with endpoints overlapping (L1, L2) and (L2, L3) and examined both the maximal adjusted P value and top 10 P values among the contacts anchored in (L1, L2) and (L2, L3). As **Supp. Table 2** shows, HiC-DC correctly identifies all of the significant events and does not assign significance to any of the contacts overlapping the control interaction bin (L2, L3).

3. What's the reproducibility of HiC-DC? If we apply HiC-DC to Hi-C data with biological replicates, can we get similar peak calling results? Similarly, if we apply HiC-DC to Hi-C data with different enzymes (HindIII vs. NcoI vs. Mbol) or different protocols (in situ Hi-C vs. dilute Hi-C), can we get reproducible peak calling results?

To address whether HiC-DC peaks calls are reproducible for different Hi-C protocols or different restriction enzymes, we chose two dilution Hi-C datasets but using two different restriction enzymes in mouse embryonic stem cells by Dixon et al. (2012). For this analysis, we trained HiC-DC on Hi-C read count data for these two restriction enzymes using a uniform 50kb binning due to the lower overall coverage of these datasets. A scatterplot of $-\log_{10} P$ values assigned to interaction bins for two experiments shows good reproducibility (**Supp. Fig. 10**, Spearman $\rho = 0.66$).

We also investigated whether HiC-DC event calls are reproducible across biological replicates. For this analysis, we used both the primary and secondary biological replicates from GM12878 provided by Rao et al. We trained our version of HiC-DC for a uniform grid on both replicates at a 5kb resolution genome-wide. Our results showed that HiC-DC $-\log_{10} P$ values are concordant between replicates for all chromosomes (**Supp. Fig. 9**, Spearman $\rho = 0.52$).

4. Page 10, line 252: the authors found distinct classes of interactions are enriched at different length scales. I am curious whether such finding is affected by TAD structure. It is known that intra-TAD interactions are much higher than inter-TAD interactions. The authors can divide all interactions into two groups: intra-TAD and inter-TAD interactions, and check whether the same conclusion holds within both groups.

We thank the reviewer for this suggestion. To address this question, we assigned TADs in B-lymphoblastoid cells based on TADs from IMR90 cells annotated by Dixon et al. 2012. Each significant interaction identified by HiC-DC was either assigned to intra-TAD or inter-TAD regions depending on how close the endpoints are to a TAD boundary. The genomic regions of significant Hi-C locus pairs are strongly enriched for genomic and epigenetic features across the two groups of TAD regions. In the **Online Methods**, we describe the details of the enrichment analysis. Inter- and intra-TAD interactions displayed different patterns of enrichments for epigenetic signals and genomic annotations. For example, significant inter-TAD interactions showed greater enrichment for DNase-DNase and K27ac-K27ac marks

at longer range genomic distances (>1.5Mb) compared to significant intra-TAD interactions (**Supp. Fig. 29**). Longer-range promoter-promoter interactions were uniquely enriched within inter-TAD regions (**Supp. Fig. 30**).

5. Page 13 line 350. The authors mentioned the advantage of HiC-DC over the popular normalization method ICE. I would love to see more in-depth analysis along that direction. For example, can HiC-DC achieve higher reproducibility of normalized Hi-C data between biological replicates than ICE? Can HiC-DC achieve better effect of bias removal than ICE? One possible analysis is to check whether normalized Hi-C data, by HiC-DC or ICE, is still affected by local genomic biases. Heatmaps similar to Yaffe and Tanay 2011 Figure 1d, 1f and 1h will be very helpful.

HiC-DC was developed to model the background distribution of Hi-C bin counts, which is a byproduct of the biases inherent in the assay; and then to assign a P value for an observed bin count comes relative to a null distribution. Although HiC-DC is not designed as a normalization procedure such as Yaffe and Tanay's method or ICE, we can use our method to normalize the Hi-C contact matrix by dividing the observed bin count by the expected bin count estimated from the model (O_{ij}/E_{ij}) and then determine the extent to which HiC-DC controls for systematic biases in the data across replicates or protocols. Following Yaffe and Tanay, we computed a heatmap of normalized counts (O_{ij}/E_{ij}) as a function of GC content in bin i and bin j for mouse ES Hi-C data generated with the HindIII and with the NcoI enzymes (**Supp. Fig. 11**). Similar to findings reported by Yaffe & Tanay 2011, we observed that there are preferential Hi-C contact patterns among low regional GC content and high regional GC content, but these patterns were largely consistent between the two restriction enzymes (**Supp. Fig. 11**).

6. Page 15 line 411. The authors mentioned that when applying HiC-DC with a uniform binning of the genome, they add an additional covariate of effective sequence space. I wonder whether such effective sequence space is also useful in non-uniform binning? When merging every 10 RE fragments into interval, different intervals can still have different effective sequence spaces. In addition, according to Yaffe and Tanay 2011 Figure 1c and 1d, the unbalance fragment size leads to ligation efficiency bias. The author may also explore such fragment bias in HiC-DC with non-uniform binned Hi-C data.

We thank the reviewer for this suggestion. The effective sequence space associated with a uniform bin is the sum of regions 500bp from restriction fragment ends (i.e. the union regions from which reads are counted) within a particular bin. This use of "effective sequence space" was previously used in HiCNorm (Hu et al., *Bioinformatics* 2012). Aggregating consecutive restriction fragments into intervals already normalizes for effective sequence space; this approach is used by Fit-Hi-C (Ay et al., *Genome Research* 2014).

As pointed out by the reviewer, Yaffe and Tanay found a fragment length bias for *cis* interactions, presumably due to differences in ligation efficiency. However, compared to the Hi-C data sets available in 2011 when this analysis was published, newer Hi-C data sets incorporate many improvements to the protocol and are much deeper. To ask whether there still exists a fragment size bias in high-resolution Hi-C data, we performed the following analysis for the non-uniform binning based on RE fragments for the Rao et al. data on chromosome 1. First, we plotted the observed bin counts for the top and bottom 10% of the bin length distribution as a function of genomic distance; next, we plotted the top and bottom 10% percentiles of expected bin counts as estimated by the HiC-DC model (**Supp. Fig. 12**). While the observed counts for the 10% longest and shortest bins do slightly deviate slightly from each other as a function of genomic distance, with marginally higher counts for the shortest bins, this variation is small compared to the difference between the top and bottom 10% percentiles as estimated by the model. Therefore, while there is some variation due to bin length size that is not accounted for in the model, the amount of this variation is very small compared to the variation due to the modeled covariates (GC content and mappability).

As a second analysis to examine this question, we compared the P values assigned to significant event called by HiC-DC on non-uniform and uniform binned Hi-C data. In order to define a reasonable correspondence between these different binnings, we found the set of overlapping bins and restricted to pairs with at least 85% overlap in area with the non-uniform and uniform bin and required the overlapping intervals to have length at least 3kb. This analysis confirmed that the $-\log_{10} P$ values for significant events from the two models were well correlated (**Supp. Fig. 13**, Spearman $\rho = 0.71$).

7. Page 16, line 437: *The authors used dispersion parameter α to account for over-dispersion in negative binomial distance. Since Hi-C data show high heterogeneity, does the constant α fit data well? Can they allow α to be a function of 1D genomic distance? Are their different dispersion parameters for intra-TAD and inter-TAD interactions?*

We thank the reviewer for this suggestion. To test whether the dispersion parameter might vary as a function of genomic distance, we binned the Hi-C data by genomic distance in 25kb bins and trained a separate hurdle negative binomial regression model on each bin. We found that the dispersion values estimated in this way do vary somewhat genomic distance (**Supp. Fig. 34**). In particular, at genomic distances of >1Mb, the estimated dispersion parameter is close to 0, suggesting that the bin count distribution is closer to Poisson at this distance. There is also a near-monotonic decrease in the estimated dispersion from 500kb to 1Mb. However, there are also signs of overfitting using this simple binning approach, visible in the scatter of dispersion values estimated for adjacent bins, since we are no longer training on all the data and using a spline to fit the dependence on genomic distance.

Therefore, while there is evidence that allowing the dispersion parameter α to vary as a function of genomic distance would produce a better null model, it would be important to develop the appropriate parametric form to describe $\alpha(d)$, and it would also be necessary to extend the underlying GLM optimization routines. We believe these matters could be appropriately addressed in future work but are out of scope for the current manuscript; we now allude to this future extension in the **Discussion**. We also wanted to confirm that the use of a constant versus distance bin-dependent dispersion values do not alter HiC-DC's significant interaction calls too severely. **Supp. Fig. 35** plots the $-\log_{10} P$ for the common dispersion model (x-axis) versus the individual dispersion models (y-axis) for different ranges of genomic distance. The single-dispersion model produces somewhat more conservative P values at all distance ranges, but there is high correlation of P values between the two approaches, and all the significant interactions reported by our single-dispersion HiC-DC model would also be reported as significant with distance-dependent dispersion parameters.

For the final point, since intra- vs inter-TAD interactions bins have different genomic distance distributions, it is therefore likely that they would have different dispersions based on above analysis. We do mention the idea of TAD-aware estimates of dispersion parameters in the **Discussion**.

8. Figure S3 and S4: *It is interesting to see Fit-Hi-C provides P-value with inflated significance compared to HiC-DC. Is it because the zero-truncated negative binomial model used in HiC-DC fits the data better than binomial model used in Fit-Hi-C? The authors may provide some numerical evidence of model fitting.*

Fit-Hi-C uses a spline to estimate the binomial probability $p(d)$ as a function of genomic distance d . The number of trials N (number of Hi-C read pairs with linear distance d) and $p(d)$ are parameters of binomial distribution. When N is large, the binomial distribution approximates a Poisson distribution. We performed a statistical test based on a Poisson regression introduced by Cameron and Trivedi (1990) to test the null hypothesis that dispersion parameter is equal to zero versus the alternative hypothesis that dispersion factor is greater than zero. We trained the Poisson regression version of the HiC-DC model on GM12878 data at 5kb binning for 23 different chromosomes and used the `dispersiontest` function in the R package AER, which implements the Cameron and Trivedi test for overdispersion. After using a Bonferroni correction for testing for all chromosomes, we can reject the null hypothesis that the dispersion is 0 with $P < 1.51e-16$. Overall, this analysis provides additional statistical evidence that the failure to model non-

zero dispersion in Fit-Hi-C as well as Poisson regression models contributes to inflation of P values (as suggested by the Q-Q plots in **Fig. 1b,c** of the main manuscript).

Minor comments:

1. I cannot find Figure 1. In the combined PDF file, page 24 is blank. The authors need to add Figure 1 in the revision.

We apologize – this problem apparently happened for some reviewers but not others. We have reduced the file size and hope this problem is fixed now.

2. The authors can add some description of computational details of HiC-DC. What's the memory requirement and time cost of using HiC-DC to conduct genome-wide peak calling on the high resolution Hi-C dataset?

HiC-DC can be run separately for each chromosome in parallel. For example, using 8 cores, it takes 13, 7, and 3 minutes to run HiC-DC chromosomes 1, 11, and 22 (i.e. examples of large, medium, and small chromosomes), respectively. The memory requirements for these runs are respectively 14.2, 9, and 5 GB for chromosomes 1, 11, and 22.

3. Page 6, line 138. HiC-DC trains the ZTNB model using a sample of 1% of all interactions. The authors may provide more details on how to sample this 1%? Is ZTNB model and HiC-DC result sensitive to different 1% samples? Some sensitivity analysis and robust test will be more convincing.

To address the question of stability of the model, we first trained the model once on a 1% sample and took the significant interactions at a fixed FDR threshold as our “true” interactions; then we asked whether these same interactions are correctly identified as we retrain on different 1% samples of the data (i.e. the power of the models fit on different training samples to detect the “true” interactions). We performed this analysis on chromosome 11 at 5kb resolution for the Rao et al. dataset. We chose bin counts with an adjusted P value below the 1% FDR threshold as our “true” interactions events for the initial model, then we trained HiC-DC on 100 different samples of the training data and computed the corresponding adjusted P values for all Hi-C locus pairs. The output of the sampling experiment is a matrix of adjusted P values for each interaction bin across the different models. We extracted a submatrix consisting only of the “true” interaction bins and calculated the average power of the models for detecting each true interaction bin, i.e. the fraction of models that detect the interaction **Supp. Fig. 8** shows the cumulative distribution (over “ground truth interactions”) of the fraction of times each “true interaction” is detected. We found that 95% of ‘true interactions’ were detected at 1% FDR by at least 90% of the models, and 90% of the interactions were detected by 100% of the models, showing that the significant contacts are not sensitive to different 1% training samples.

4. Can we apply HiC-DC to normalize inter-chromosomal interactions and identify inter-chromosomal interaction peaks?

It should be possible to use a simplified version of HiC-DC to estimate a null model for inter-chromosomal Hi-C bin counts and therefore assess the significance of inter-chromosomal interactions. For each pair of chromosomes, we would use the matrix of bin counts between pairs of intervals as output values in the hurdle negative binomial regression model; the covariates of the model would be GC content, mappability, and effective sequence space. The dependence of random polymer ligation on linear genomic distance only makes sense in the intra-chromosomal setting and would not be incorporated here. While our focus in this paper was intra-chromosomal interactions and integration with epigenomic data, we hope to return to inter-chromosomal interactions in future work, and we now mention this potential extension in the **Discussion**.

5. In Figure S10 and S11, it is better to also show raw Hi-C contact matrix and HiC-DC results, similar to Figure S9a, S9b.

In response to this suggestion, we have included both the raw Hi-C contact matrix and the matrix of $-\log_{10}$ P values generated by HiC-DC in **Supp. Fig. 21** and **22**.

Reviewer #3 (Remarks to the Author):

In this work, Carty et al. developed an innovative statistical model (HiC-DC) to detect significant interactions in Hi-C data. This model is based on a hurdle negative binomial regression that takes consideration of many known biases in Hi-C interaction matrices, such as GC content, mappability, and the distance dependence effect due to random polymer collision. The innovation of this proposed method is that the authors addressed the zero-inflation and overdispersion, which are commonly seen in Hi-C data, but have not been effectively addressed in other existing Hi-C analysis software.

Overall, the manuscript is well written and the analysis was properly done. The figures are clear and precise. The software performed well and the applications to high-resolution Hi-C data also lead to some interesting discoveries regarding gene regulatory activities, such as different structural and regulatory interactions are enriched in distinct distance ranges. The work will provide a great tool for the 3D genome organization community and novel biological findings as well.

We thank the reviewer for this positive assessment of our manuscript.

That being said, I do have some comments and suggestions that I hope the authors can address:

1. Overall, the authors did a great job reviewing the previous efforts and the remaining challenges in analyzing and modeling Hi-C data. I do suggest the authors reviews a couple of other Hi-C peaking calling methods, namely, HiCCUPS by Rao SS et al 2014, and Zheng Xu 2015 et al.

We thank the reviewer for the suggestion. We have now included a brief description of both the HiCCUPS algorithm described in Rao et al. (2014) and the hidden Markov random field method of Xu et al. (2015) in the **Introduction**. We also include more detailed comparisons of our HiC-DC method to HiCCUPS, as requested by Reviewer #1 and described above.

2. Line 158: the authors' claim that there is P-value inflation of by Fit-Hi-C need to be further justified. More details of how this figure was generated and more discussion will be helpful.

We first want to acknowledge that there is no gold standard for “true” 3D chromosomal interactions – other 3C-based methods like ChIA-PET have their own sources of bias that are not well understood, and examining enrichments of epigenomic annotations as we do in the manuscript provides indirect evidence that called interactions may be functional. We try to be careful in the manuscript only to claim that there is evidence for P value inflation in several other methods that do not account for overdispersion of read count data, including Fit-Hi-C and variants of the HiC-DC model where we use Poisson regression rather than negative binomial regression, rather than claiming that these methods call false positives. We use the Q-Q plot analysis (described more in the next paragraph) as a way of demonstrating that P values can be “tamed” by using appropriate choices for modeling the read counts in the null model. We have added a brief note to the Discussion to acknowledge these caveats and avoid overclaiming.

Q-Q plot analysis is a standard approach for assessing P value inflation that has been used for example in the GWAS literature. The assumption we are making here is that most read count bins do not represent “true” interactions but rather can be described by the null model, based on systemic sources of bias and on random ligation effects that depends on linear genomic distance. If the null model correctly

describes the background distribution of non-interactions, then the P values produced by model should follow a uniform distribution. We construct a quantile-quantile (Q-Q) plot of $-\log_{10} P$ -values from the model versus $-\log_{10} P$ -values generated from a uniform distribution to verify that most of read count bins indeed follow the null model. If the Q-Q plot quickly deviates from the $y=x$ line, it means that many data points are being assigned very small P values by the null model and that likely the null model is not correctly accounting for variation in the background model. We suspect that because Fit-Hi-C uses a binomial distribution (where $p(d)$ is the binomial probability parameter that is estimated using B-spline) which does not account for overdispersion in read count data, its P values tend to be too small, as suggested by the Q-Q plot (**Fig. 1c**). Consistent with this hypothesis, if we replace negative binomial regression with Poisson regression – i.e. if we assume no dispersion in the read count data – in the HiC-DC model, then HiC-DC P values also appear to be inflated (**Fig. 1b**). We have added a few sentences to the **Results** section to explain this reasoning.

3. The authors continued to argue that HiC-DC reported much less significant interactions than Fit-Hi-C (1 million vs. 3 million). Can the authors draw a Venn diagram and show where the differences are? Further, for the differential peaks, can they compare with other types of data (say ChIA-PET to get an evaluation)?

We first want to clear up some inconsistencies in how we originally reported the numbers of significant interactions for HiC-DC and Fit-Hi-C. In order to compare to multiple methods for the revision, we computed HiC-DC P values on the Rao et al. GM12878 primary replicate at different resolutions (uniform and non-uniform binnings) and now report the number of significant interactions at 5% and 1% FDR (total and by chromosome) in **Supp. Table 1**. We report these numbers both including and excluding the diagonal interaction bins ($D=0$) bins; the $D=0$ significant ‘interactions’ may represent regions of increased accessibility that lead to higher local ligation events. We also report the number of interactions reported by HiCCUPS as well as the 5% and 1% FDR numbers for different versions of Fit-Hi-C in **Supp. Table 1**. We realized that Fit-Hi-C results change dramatically if ICE is not used, with far fewer significant interactions post-ICE normalization, and more modestly if read quality filtering is used, with somewhat fewer significant events with the MAPQGE30 map compared to the MAPQG0 map. Previously, we reported 5% FDR numbers for Fit-Hi-C pre-ICE (>3M significant interactions) and for HiC-DC including $D=0$ interactions (>1M significant interactions). Since Fit-Hi-C filters out $D=0$ interactions and since only its post-ICE results are comparable to HiC-DC, we now report 1% FDR numbers for Fit-Hi-C post-ICE (~793K interactions) and for HiC-DC excluding $D=0$ interactions (~321K interactions). We believe these numbers are the most relevant and comparable for the HiC-DC vs. Fit-Hi-C analysis.

The Venn diagram corresponding to this comparison of Fit-Hi-C with HiC-DC is in **Supp. Fig. 6**; the dependence of Fit-Hi-C’s significant interactions and overlap with HiC-DC on ICE normalization and MAPQG0 vs. MAPQGE30 is shown in **Supp. Fig. 7**.

;

With respect to the Reviewer’s main question about comparison with other types of data like ChIA-PET on the differential peaks, we did not perform this specific comparison because the proper statistical analysis of ChIA-PET data is also an area of active research with no accepted solution, and at least some published data sets have limited reproducibility between replicates. We do agree that as these data become more standard and reproducible, it would be important to validate Hi-C interaction calls with other types of 3C-based experimental data. In the meantime, we are careful to acknowledge that there is no gold standard for 3D interaction data, that the enrichment analysis we have performed is suggestive but not conclusive evidence of correct P value estimation, and that the best evidence we have that failure to model overdispersion leads to P value inflation is the comparison of Poisson and negative binomial versions of the HiC-DC model on the same data set (as described above in response to the previous comment).

4. It is interesting that the authors discovered different classes of interaction are enriched in different length scales. For example, the CTCF-mediated interactions are most enriched in the distance range 700Kb -1.5Mb. However, the underlying biological implication of these differences is not fully discussed.

We believe that the enrichment of CTCF-associated interactions at the 700kb-1.5Mb distance range coincides with DNA loops that define the boundaries of topologically associated domains (TADs), which are ~1Mb in length on average (Dixon et al., 2012). We have added this potential explanation to the text.

5. Figure 3. What is the rationale to plot "DNase-K27ac to K27ac", "CTCF-DNase-K27ac to DNase-K27ac"? Since GM12878 is an ENCODE cell line, why not just use their definition of enhancers, promoters etc.

For the main results presented in the text, we use epigenomic data from GM12878 rather than predicted chromatin states because we believe it more direct and accurate to use the original data. For example, while the Segway + ChromHMM annotations available for GM12878 from the UCSC genome browser site include a CTCF state, this state does not fully coincide with actual reproducible CTCF peak calls in GM12878 (data not shown) – at least, each genomic interval may overlap with multiple predicted states, and it is unclear how to prioritize states that have inferred rather than concrete meanings.

Nevertheless, we agree with the reviewer that it can be informative to perform the enrichment analysis with widely used predicted chromatin states. Therefore, we took the Segway + ChromHMM genome browser annotations for GM12878 (segmentation of the genome into 7 'merged' states based on 25 distinct chromatin states used by the two methods) as described above and performed Fisher's exact test enrichment analysis for these states. These results are now included in **Supp. Fig. 25 and 26** and recapitulate some of our major findings, such as the enrichment of distal promoter-promoter interactions. We note that only 4% of genomic bins previously annotated as containing CTCF binding sites based on ChIP-seq data were assigned a "CTCF" state in this analysis – possibly because the CTCF states correspond to short segments and states with longer states (e.g. "Repressed") dominate the annotations. However, since there is no straightforward way to prioritize annotations other than length of overlap, we left the analysis as is and flag this issue in the text.

6. Figure 1a, the labels for the blue line and green line are inconsistent with description in its figure legend.

We apologize for the confusion and have corrected the labels for **Fig. 1a**.

Reviewers' Comments:

Reviewer #3 (Remarks to the Author)

The authors have addressed all my concerns.